# Neuropathology of Animal Prion Diseases

**DOI:** 10.3390/biom11030466

**Published:** 2021-03-21

**Authors:** Leonor Orge, Carla Lima, Carla Machado, Paula Tavares, Paula Mendonça, Paulo Carvalho, João Silva, Maria de Lurdes Pinto, Estela Bastos, Jorge Cláudio Pereira, Nuno Gonçalves-Anjo, Adelina Gama, Alexandra Esteves, Anabela Alves, Ana Cristina Matos, Fernanda Seixas, Filipe Silva, Isabel Pires, Luis Figueira, Madalena Vieira-Pinto, Roberto Sargo, Maria dos Anjos Pires

**Affiliations:** 1Animal and Veterinary Research Centre (CECAV), Associate Laboratory for Animal and Veterinary Science–AL4AnimalS, University of Trás-os-Montes and Alto Douro (UTAD), 5000-801 Vila Real, Portugal; lpinto@utad.pt (M.d.L.P.); jorgecpereira599@gmail.com (J.C.P.); agama@utad.pt (A.G.); alexe@utad.pt (A.E.); aalves@utad.pt (A.A.); fseixas@utad.pt (F.S.); fsilva@utad.pt (F.S.); ipires@utad.pt (I.P.); mmvpinto@utad.pt (M.V.-P.); roberto.sargo@gmail.com (R.S.); 2Pathology Laboratory, UEISPSA, National Institute for Agricultural and Veterinary Research (INIAV), I.P., 2780-157 Oeiras, Portugal; carla.neves@iniav.pt (C.M.); paula.mendonca@iniav.pt (P.M.); paulo.carvalho@iniav.pt (P.C.); joao.silva@iniav.pt (J.S.); 3Pathology Laboratory, UEISPSA, National Institute for Agricultural and Veterinary Research (INIAV), I.P., 4485-655 Vairão-Vila do Conde, Portugal; carla.lima@iniav.pt (C.L.); paula.tavares@iniav.pt (P.T.); 4Centre for the Research and Technology of Agro-Environmental and Biological Sciences (CITAB), Associate Laboratory Institute for innovation, capacity building and sustainability of agri-food production-Inov4Agro, University of Trás-os-Montes and Alto Douro (UTAD), 5000-801 Vila Real, Portugal; ebastos@utad.pt (E.B.); nuno_anjo@hotmail.com (N.G.-A.); 5Research Center for Natural Resources, Environment and Society (CERNAS), Polytechnic Institute of Castelo Branco (IPCB), Av. Pedro Álvares Cabral, 12, 6000-084 Castelo Branco, Portugal; acmatos@ipcb.pt; 6Quality of Life in the Rural World (Q-Rural), Polytechnic Institute of Castelo Branco (IPCB), Av. Pedro Álvares Cabral, 12, 6000-084 Castelo Branco, Portugal; lmftfigueira@gmail.com

**Keywords:** prion, neuropathology, spongiform degeneration, animal TSE, neuroinflammation, gene *PRNP*

## Abstract

Transmissible Spongiform Encephalopathies (TSEs) or prion diseases are a fatal group of infectious, inherited and spontaneous neurodegenerative diseases affecting human and animals. They are caused by the conversion of cellular prion protein (PrP^C^) into a misfolded pathological isoform (PrP^Sc^ or prion- proteinaceous infectious particle) that self-propagates by conformational conversion of PrP^C^. Yet by an unknown mechanism, PrP^C^ can fold into different PrP^Sc^ conformers that may result in different prion strains that display specific disease phenotype (incubation time, clinical signs and lesion profile). Although the pathways for neurodegeneration as well as the involvement of brain inflammation in these diseases are not well understood, the spongiform changes, neuronal loss, gliosis and accumulation of PrP^Sc^ are the characteristic neuropathological lesions. Scrapie affecting small ruminants was the first identified TSE and has been considered the archetype of prion diseases, though atypical and new animal prion diseases continue to emerge highlighting the importance to investigate the lesion profile in naturally affected animals. In this report, we review the neuropathology and the neuroinflammation of animal prion diseases in natural hosts from scrapie, going through the zoonotic bovine spongiform encephalopathy (BSE), the chronic wasting disease (CWD) to the newly identified camel prion disease (CPD).

## 1. Introduction 

Even though the histopathological examination is a less sensitive diagnostic method for transmissible spongiform encephalopathies (TSEs) compared to detection by immunohistochemistry, PETblot and Western blot of PrP^Sc^, the prion diagnostic marker, it is indispensable in the study of the lesions in TSEs to identify and characterize TSE phenotypes in both natural hosts and experimental animals. 

In TSEs, there are no gross neuropathological lesions but there are characteristic histological lesions: bilateral and symmetrical widespread neuropil and/or neuronal vacuolation (spongiform appearance), synaptic changes, neuronal loss, gliosis, a variable degree and type of accumulation of PrP^Sc^ and sometimes amyloid plaques [1,2,3] (Figure 1). Spongiform changes are a general hallmark of TSEs; albeit neuronal vacuolation is frequent in animal prion diseases, it is uncommon in humans and has to be distinguished from non-specific spongiosis. This includes brain oedema, metabolic encephalopathies, autolysis and artefacts [1]. TSE spongiform degeneration is widespread or focal with small or oval uniform-sized, well-delineated and empty vacuoles in the neuropil and the neuron perikaryon. These changes can sometimes result from an artefact caused by the fixation and paraffin-embedded processing of the brain tissue [4]. It might also be caused by abnormal membrane permeability, autophagy or accumulation of PrP^Sc^ in the lysosomes [4]. Some studies revealed that vacuolation is not related to PrP^Sc^ accumulation [4].

Gliosis and PrP^Sc^ deposition, not always associated with the severity of lesions, precede neuronal damage and neuropil vacuolation in the brain suggesting that a neuroinflammatory component might contribute to neuronal loss in these diseases [5]. Nevertheless, the mechanism of neuronal damage, the precise role of microgliosis and astrogliosis in that neurodegeneration and the cell tropism by a specific prion strain require additional research. 

Although these lesions are quite consistent in TSEs, prion strains originate different distribution of the lesions in the central nervous system (CNS) (lesion profiles) as well as different morphological and biochemical profiles of the PrP^Sc^ deposition in a certain host [2] that have to be confirmed definitely by mouse bioassays. As this complete strain characterization is not possible in every TSE case naturally identified, it is crucial to perform a systematic neuropathological analysis in each disease and each natural host [3], contributing to a better pathogenesis understanding of prion diseases. 

For this review, the lesion profiles obtained after experimental transmission of a TSE to wild type or transgenic rodents will not be discussed. Thus, the present work compiles and compares the neuropathological lesions described in the original animal species affected by prion diseases. These include the two naturally acquired TSEs (scrapie of small ruminants and chronic wasting diseases affecting wild and captive cervids), the acquired TSEs related to prion contaminated feed (transmissible mink encephalopathy and the zoonotic bovine spongiform encephalopathy (BSE)), the TSEs linked to the BSE epidemic (feline spongiform encephalopathy of zoological and domestic cats (FSE) and TSE of zoological ruminants and non-human primates) and the putative sporadic diseases (atypical scrapie, H-BSE, L-BSE and camel prion disease). 

## 2. Prions and Animal Prion Diseases 

Prion is defined as a “proteinaceous infectious particle” [6] consisting exclusively of a single protein without the involvement of nucleic acids that causes spongiform encephalopathies in mammals. Prion diseases are characterized by the accumulation of abnormal isoforms of PrP glycoprotein. Disease-associated isoform, designated PrP^Sc^ when protease-resistant, is derived from its physiological cellular prion protein precursor PrP^C^, by a posttranslational mechanism that involves conformational change and aggregation [7]. The pathogenic misfolded isoform is able to transmit misfolding by transforming more PrP^C^ towards an increasing in β-sheet structure in detriment to the physiological α-helical structure and a tendency to aggregate into oligomers [8].

The prion protein gene (*PRNP*), highly conserved across mammals, is considered the major genetic determinant of susceptibility to prion diseases. Exon composition varies from one exon (e.g., *Canis lupus familiaris*, *Felis catus*, and *Ovis aries*), two exons (*Homo sapiens*), three exons (*Cervus elaphus*) or four exons (like *Bos taurus),* according to the Ensembl database. Nevertheless, a high level of conservation is observed in the coding sequence and aminoacidic sequence. A comparative genomic approach including nine species allows us to visualize the more conserved regions of the protein, as presented in Figure 2.

Multiple mutations and polymorphisms of *PRNP* have been described in different animal species and some of them demonstrated an important effect on prion disease susceptibility/resistance (Table 1). Figure 3 presents the missense mutations detected in PrP protein from *Ovis aries*. Variants are classified above and below the bar as ‘Risk factor’ and ‘Resistance’ according to in vivo and in vitro analyses [11,12,13,14]. The other variants were analysed by an in-silico approach through the PROVEAN algorithm (http://provean.jcvi.org/index.php, accessed on 27 January 2021) and classified as ”Deleterious” and ”Neutral”. All the missense mutations were gathered from Ensembl (ENSOART00020017839.1) and through database searching [13,14,15,16,17,18].

Different prion strains (isolates or variants) are distinct self-templating conformers [33] that display different phenotypes in a specified host and are preserved upon serial passage within the same host genotype. These distinct entities were first proposed in 1961 and since then have been characterized in different mammalian species with the implication in disease pathology and transmission [34]. Prion strains can be differentiated after experimental transmission to wild type or transgenic rodents by differences in the clinical signs, incubation period and the lesion profiles in the brain of the affected animals leading to the laboratory definition of a strain [3]. The lesion profile mainly focuses on differences in the regional patterns of prion-induced vacuolar neuropathology and/or PrP^Sc^ deposition but there are also strain differences in the association of PrP^Sc^ with particular brain cell types [35]. At the same time, biochemical characteristics of PrP^Sc^ as the electrophoretic mobility, protease resistance, glycosylation profile and sedimentation can be used to distinguish between prion strains [36].

As various distinct phenotypes can be identified in mice, the occurrence of different prion strains is known in classical scrapie (CS) of sheep and goats (revised by [37]), the first TSE described nearly 300 years ago, in a rare TSE in mink farms (TME) and chronic wasting disease (CWD) affecting farmed and free-ranging cervids, both identified in North America in 1947 and 1967, respectively (revised by [38]) (Figure 4). Contrarily, single strains have been associated with other animal prion diseases such as classical Bovine Spongiform Encephalopathy (C-BSE). C-BSE, responsible for an epidemic and a public health crisis in Europe, was identified in 1985 and largely increased the interest in the TSEs research due to its link to the variant Creutzfeldt-Jakob disease (vCJD) in humans and the putative transmission to other animal species (felids, zoo ruminants and small ruminants) [39]. This research and the reinforced TSE surveillance programs lead to the identification of more animal TSEs. As they showed distinct features from any TSE cases known at the time, the classical forms, they were designated as atypical (revised by [38]): atypical scrapie, atypical BSE and atypical CWD. Recently, a new prion disease was also confirmed in dromedaries—camel prion disease (CPD) (Figure 4). These could be new animal prion diseases or diseases not previously detected. Retrospective studies indicated that atypical scrapie has been present since at least 1972 [40]. 

## 3. Neuropathology 

### 3.1. Scrapie

Classical scrapie (CS), the ancient animal transmissible spongiform encephalopathy affecting sheep and goats, has been the prototype model to study the pathogenesis and neuropathological changes observed in TSEs. In 1998, a different form of scrapie (named Nor98 or atypical scrapie, AS) was identified in Norway [41,42]. The two scrapie forms differ in epidemiology, affected *PRNP* genotypes, clinical presentation, lesion profile and type and distribution of PrP^Sc^.

#### 3.1.1. Classical Scrapie

Neurological lesions depend on scrapie strain and host *PRNP* genotype but mostly include vacuolation of neuronal perikarya and grey matter neuropil, neuronal degeneration, gliosis (predominantly astrocytic) and amyloidosis detected mainly in diencephalon, midbrain, pons, medulla oblongata and cerebellar cortex [43,44]. Vacuolation of neuronal perikarya is not pathognomonic and it can be observed in brains of apparently healthy sheep; in these instances, however, the number of vacuoles is typically much smaller than in cases of clinical scrapie (reviewed by [45]).

Several different prion strains have been associated with the development of classical scrapie. First identified in UK, France and Italy, those showed different biological features based on the serial passaging of natural isolates through bioassay experiments in rodents (reviewed by [37]).

Regarding the sheep genotype, [46] reported the variability in the sheep *PRNP* gene with three major polymorphic positions at codons 136, 154 and 171. Different authors confirmed the association of the genotypes with the risk of developing classical scrapie. At codon 136, valine (V) was associated with scrapie susceptibility, while alanine (A) was associated with resistance [47,48]. At codon 154, arginine (R) was associated with susceptibility, while histidine (H) was associated with resistance [49]. At codon 171, glutamine (Q) and histidine (H) were associated with susceptibility, while arginine (R) was associated with resistance [50]. Combinations of these polymorphisms could result in twelve alleles. Nevertheless, only five commonly seen haplotypes or alleles (ARQ, AHQ, ARR, ARH, VRQ) were frequently found in sheep. Although sheep with the ARR/ARR genotype are highly resistant to the scrapie agent, there are some reports of naturally occurring classical scrapie disease in sheep presenting that genotype [51]. Despite these three codons having a strong association with CS, codon 171 has a greater influence compared to the other two codons, where homozygous QQ sheep are susceptible to scrapie and the RR genotype confers resistance (Table 1) [13]. In 2003, a novel variation was detected in codon 171 (lysine-K) by Guo et al. (2003) [52] and later, Boukouvala et al. (2018) [53] reported that dairy sheep with K171 showed a 75% reduction in the risk of developing scrapie when compared with individuals with Q171. Recently, Cassmann et al. (2019) [14] showed that individuals homozygous for K171 (ARK/ARK) were resistant to CS after oronasal inoculation with PrP^Sc^. Cassmann & Greenlee (2020) [54] suggest that probably the influence of K171 polymorphism on resistance to scrapie is greater when compared to Q171 polymorphism but minus when compared to R171 polymorphism.

Histopathological examination, historically based on examination of a single section of medulla oblongata taken at the level of the obex, (the earliest consistent neuroanatomical site for morphological vacuolar changes, mainly at the dorsal vagal nucleus (DVM)) [43] is still valid for the confirmation of CS [55]. Although, according to Wood and colleagues (1997) [43] the distribution of vacuolation in the grey matter neuropil could be classified into seven patterns (I, IIa, IIb, IIIa, IIIb, IIIc, IV) based on location and severity of neuropil vacuolation throughout the encephalon. All of them show vacuolar lesions in the brainstem and in the cerebellum, but there was some variation, especially in the telencephalon and hypothalamus, where lesions were not always present [43]. Besides the DVM, at the level of pons and obex—the papillioform nucleus of the pons, the pontobulbar reticular formation, the midline raphe, and the lateral cuneate and lateral caudal nuclei are the most consistently affected as well as substantia nigra in the midbrain [43]. In the cerebellum, the lesions are most severe in the lingula. Neuropil vacuolation is sometimes severe and diffuse in the granular layer, extending into the molecular layer [43].

However, as detectable PrP^Sc^ precedes vacuolation and clinical signs, the immuno-based analysis is a more sensitive tool and, currently, a diagnostic procedure. Clinically suspect cases of scrapie should (if suitable samples are available) continue to be investigated initially by histopathological examination for morphological changes, but diagnostic criteria must include the demonstration of PrP^Sc^ in the central nervous system (CNS) by immunohistochemistry (IHC) and/or Western immunoblotting (WB). The medulla oblongata is the most consistent and appropriate diagnostic region of the CNS for CS [55] (Figure 5).

The application of IHC in the sheep encephalon revealed a variety of different intra- and extracellular PrP^Sc^ deposits (PrP^Sc^ types) identified and classified morphologically by different authors, despite differences in nomenclature and in the number of PrP^Sc^ types recognized [44,56,57,58]. These PrP^Sc^ types can be grouped into PrP^Sc^ patterns in order to facilitate comparisons between the affected sheep and to define PrP^Sc^profiling [58,59]. Nevertheless, this grouping approach could miss PrP^Sc^ types distribution in specific neuroanatomical nuclei in a certain encephalon region resulting in a significant loss of discriminatory potential of prion strains [44]. 

Based on several studies [44,58,60], the following PrP^Sc^ types were established (Figure 1; Figure 6):

Intracellular:Intraneuronal—fine PrP^Sc^ granular deposits to coarse, sometimes confluent scattered in the cytoplasm of the perikarya of neurons surrounding the nucleus;Intraglial—intense granular or ovoid deposits of PrP^Sc^, slightly larger than those observed in neurons, in close proximity to the glial cell nucleus.

Extracellular:Stellate (also named glial)—radiating, branching PrP^Sc^ deposits centred on a visible glial-type nucleus, conferring a star-like appearance;Perivascular—PrP^Sc^ deposits located around the blood vessels in the white matter;Subpial—loose-mesh to more amorphous, multi-located continuous PrP^Sc^ accumulations occurred beneath the pia mater;Subependimal—similar to the perivascular PrP^Sc^ type but in the glial layer underneath the ependymal lining of the ventricular system deposits, usually discontinuous and seen mainly around the lateral ventricles;


These four PrP^Sc^ types are related to astrocyte processes.
Linear—thread-like deposits of PrP^Sc^ located in the neuropil;Fine granular (fine punctate or fine particulate)—Numerous, small PrP^Sc^ granules observed in the neuropil;Aggregates (or coarse granular, coarse particulate to coalescing and moss-like)—appear as large amorphous PrP^Sc^ accumulations scattered in the neuropil;Perineuronal—thin deposits of PrP^Sc^ around an individual, scattered neuronal perikarya and neurites;Plaques-Like (or plaques, vascular plaques)—fibrillar, radiate relatively large accumulations of PrP^Sc^ often distributed around blood vessels of a different calibre.

Primarily, the analysis of the differences in intensity and relative proportion of each of these PrP^Sc^ types indicated an effect of the scrapie strain on the PrP^Sc^ profile and a possible effect of the host *PRNP* genotype on the amount of PrP^Sc^ accumulation in the encephalon, apparently related to the incubation period [58]. These findings suggested that different scrapie strains could be distinguished using PrP^Sc^ IHC examination of the encephala of affected animals, yet the detection of prion strains in field cases of scrapie using IHC failed to support those conclusions and a strong association was seen between PrP^Sc^ types and *PRNP* genotype, especially in relation to codon 136 [44]. The observed association between PrP^Sc^ types and genotypes may, in fact, be strain-induced but in natural disease, individual scrapie strains may exhibit a genotypic tropism [44]. So, although each prion strain-host combination produces a unique IHC PrP^Sc^ signature [3], this interaction between CS strains and *PRNP* genotype turns difficult to exactly discriminate classical scrapie strains in field cases just by analyzing the phenotype of individual sheep [3]. Nevertheless, some studies disclosed that, within a single *PRNP* genotype, multiple natural sheep scrapie strains exist within Europe and could be revealed by detailed pathological examinations, which can be harmonized between laboratories to produce comparable results [59,61].

Even though not consistently in all CS affected sheep (e.g., sheep with ARR allele) [62]), PrP^Sc^ deposition has been also observed in lymphoid tissues—tonsils [63], spleen [64], lymph nodes [65], nictitating membrane as well as in muscle, placenta [66], skin [67,68], mammary gland [69], distal ileum, proximal colon [65], pancreas, heart and urinary bladder [67] (Figure 5).

Albeit with a significantly lower incidence compared to sheep, natural goat CS identified since 1942 has been recorded in both mixed herds with sheep and goat flocks (reviewed by [62]). Regarding the *PRNP* gene in goat, there are at least 50 polymorphisms [70]) but their association with resistance or susceptibility to CS is less defined when compared to sheep. Lacroux and collaborators [71] proposed that goats with the H154, Q211, and K222 polymorphisms appear to be resistant to CS after oral exposure (Table 1). In relation to susceptibility, goats with Asparagine (N) in homozygous condition at position 146 seem to present a higher risk, according to several reports [22,72]). Recently, four novel non-synonymous polymorphisms were detected in goat: G67S, W68R, G69D and R159H [18] but additional studies are needed to explore their implication in the susceptibility/resistance to scrapie. In the report of EFSA (2017) [73] a ranking of resistance to classical scrapie in goats was provided in the following order: K222 > D146 = S146 > Q211 = H154 = M142 > S127 = H143 > wild type. In a recent paper, Migliore and collaborators [74] reflected about the European Union scrapie control in goat putting in evidence that breeding for resistance is often compromised by a low frequency of resistant alleles emphasizing the importance of considering the particular situation of each country regarding the relative frequency of resistant alleles. Nevertheless, it is recognized that the genetic resistance to CS in goats should be trailed for at least one of the most recognized resistant alleles (K222, D146, and S146) (Table 1).

The vacuolar changes observed in CS affected goats is similar to that described for sheep scrapie, as well as the morphologic types of PrP^Sc^ immunolabeling, although appearing more extensive and variable in the distribution in goat scrapie [75,76,77]). This variability might be influenced by *PRNP* genotype, age, clinical stage and even the prion strain [75].

The biological diversity of prion strains causing disease in goats has been less investigated and very little knowledge has been gathered on scrapie strain diversity in goats. Recent studies have characterized several naturally TSE-affected goats, potentially identifying three types of CS in goats that could be discriminated by protease sensitivity of the N-terminus of PrP^Sc^ [78]. Nevertheless, goat CS isolates can display different patterns of geographical distribution in Europe, showing composite strain features, with strain components or sub-strains in different proportions in individual goats or tissues [37].

#### 3.1.2. Atypical Scrapie

Contrary to CS, atypical scrapie (AS) (Nor98) has a single prion strain [79] and the ARR haplotype that confers CS resistance, does not seem to have the same contribution in AS [80]. The amino acids phenylalanine at codon 141 (F141) and histidine at codon 154 (H154) in the *PRNP* gene are highly associated and considered risk factors (Table 1). [13]. The first report [41] and [81] confirmed that the AHQ allele was associated with the highest incidence of AS. An increased risk of AS has been also identified in sheep with the AF141RQ haplotype [81].

The spongiform change and PrP^Sc^ deposition in AS cases occur predominantly in the cortices of the cerebellum and the cerebrum rather than the medulla oblongata, as observed in classical sheep scrapie. Neuropil vacuolation is very prominent in the molecular layer of the cerebellar cortex, neocortex, hippocampus, basal nuclei and nucleus accumbens [41,60,82] (Figure 5).

The immunohistochemistry types of PrP^Sc^ and its distribution are also different from CS. PrP^Sc^ deposition is mild in the obex, occurring mainly in the neuropil at the nucleus of the spinal tract of the trigeminal nerve and in the white matter tracts, but more intense and widespread in the cerebellum, substantia nigra, thalamus and basal nuclei with or without vacuolation [41,60,82]. Within the cerebellum, PrP^Sc^ may be distributed across the molecular and granular layers, or it could be restricted to one or the other [83].

Few PrP^Sc^ types have been recognized: fine granular, aggregates, plaque-like, linear, perineuronal, and a distinctive immunoreactivity in the white matter (punctate and globular) (Figure 1, Figure 5 and Figure 6) [60]. Fine granular and aggregate staining types are the most common, particularly in the rostral midbrain and thalamic regions and may represent PrP^Sc^ near synapses or on distal neuronal processes [60]. The distinctive PrP^Sc^ globular type is defined as a complete or incomplete (crescent) ring- or oval-shaped deposit, which may have an irregular outline. The punctate type is characterized by smaller deposits with a shape of a small circle or elongated tear-drop with also an irregular outline thought to be associated with oligodendrocytes [60].

Notably, intraneuronal and glial associated PrP^Sc^ types are not observed, probably related to a higher proteinase-K sensitivity of the PrP^Sc^ compared to the PrP^Sc^ in CS, suggesting a greater ability of cells to digest the abnormal prion (reviewed by [13]).

The diagnostic methods fail to demonstrate PrP^Sc^ in peripheral or lymphoid tissues of sheep affected with AS [41,84] (Figure 5); however, infectivity has been demonstrated in the ileum, spleen, skeletal muscle, lymphoid tissues, and peripheral nerves by bioassay [84,85].

Atypical scrapie has also been reported in goats with a similar PrP^Sc^ electrophoretic profile to the AS affected sheep, but the distribution of the lesions is more rostral, affecting the thalamus and midbrain [86]. Analogous to sheep with AS, histidine substitution at codon 154 seems to represent a risk factor for AS in goats [23].

### 3.2. Transmissible Mink Encephalopathy (TME)

Mink belong to the family Mustelidae, along with weasels, otters and ferrets. They are divided into two species: the European Mink (*Mustela lutreola*) and the American mink (*Neovison vison*). Both European and American mink, mainly the latter, are valuable for their fur. Mink oil is also appreciated due to its utilization in some pharmaceutical products and for the protection and preservation of leather [87]. The fur industry is closely linked to the American mink and for that reason, they are raised in captivity across the world. Mink can live up to ten years, although normally in the wild they rarely live for more than three years. They are carnivorous and their diet consists of frogs, salamanders, fish, crayfish, muskrats, mice and voles. Living near the water, they often explore deep water pools for some hidden prey [87]. In the larger fur ranches, the mink population can be twice as large as the wild population, due to better nutrition as well as selective breeding for size. Together with genetic tools, producers can develop a great variety of fur colours: pure white, sapphire, pearl, blue and black [87]. Due to its importance in the fur industry and hence, its economic importance, many American mink farms were established across the world, mainly in the northern countries of North America and Eurasia. These farms have been a frequent target of animal-rights activists who advocate for their freedom and release the animals into the wild. For that reason, many American mink populations have been spread far from their indigenous territory. They can be found in some parts of Europe, namely Scandinavia, Russia, Iceland and South America [87]. Being less aggressive and adaptable, the European mink once abundant in Europe is now threatened due to the invasion of the American mink, and its population is declining.

Mink can be affected by some encephalopathies, such as Chasteks’ paralysis—thiamine deficiency leading to hemorrhagic destruction of the grey matter and metachromatic leukodystrophy, a hereditary disease due to a sublethal autosomal recessive factor (revised by [88]). In 1965, Hartsough and Burger described a rare disease noticed in 1947 as an outbreak in a mink farm in Brown County (Wisconsin, USA) reaching almost 100% fatality of the adult minks [89]. This disease revealed a spongy transformation of grey matter and it was experimentally transmitted to healthy mink. To differentiate from the other encephalopathies, Marsh, Burger and Hanson, in 1969, named it “transmissible mink encephalopathy” [88].

The disease is considered to be transmitted orally, suggesting ingestion of contaminated feed (mainly aged cattle as a primary source of fresh meat for feed production) [89], and tong wounds are thought to enhance this transmission [90]. Experimental transmission is also possible by intramuscular injection. Erosions and ulcers in mucous membranes or skin might also allow the entry of prions which can spread to lymphoid tissues including the spleen and retropharyngeal lymph nodes. Nevertheless, mink-to-mink transmission seems to be rare [90].The vertical transmission does not occur, as kits in contact with infected mothers, nursing and eating the same feed, do not develop the disease. In natural infection, the incubation period can be up to six months and the clinical phase lasts from two to six weeks [88,90].

Experimental transmission studies demonstrated no susceptibility to scrapie prions as for inoculation of minks with classical scrapie strain does not cause the disease. However, the transmission of an American TME strain into cattle resulted in a TSE with an incubation period of 18.5 months; the back passage of this bovine TME into mink through oral and intracerebral inoculation led to incubation periods of four and seven months, respectively. This is similar to what was observed when the American TME strain was inoculated, oral and intracerebrally, in minks. The pathogenicity of the American strain was not reduced or changed when transmitted into cattle, suggesting that cattle are susceptible to TME, and even that a TSE in cattle might have been the source of TME infection. Investigation in TgOvPrP4 mice by [91] demonstrated the similarities between bovine TME and L-type BSE electrophoretic profiles, survival periods and the distribution of vacuolar pathologic changes [89,91]. Commoy et al. [92] demonstrated the transmissibility of cattle-adapted TME to macaques (similar to L-type BSE) and humanized transgenic mice (the secondary passage to transgenic mice expressing bovine PrP maintained the strains used in the study). These results are important as they suggest a possible zoonotic potential of TME and this may impact human health [92]. Experimental transmission of TME to raccoons was also shown [93].

TME-infected minks present intense neuropil vacuolation in the cerebral cortex (frontal cortex), thalamus, hypothalamus and corpus callosum and a less severe spongiosis in the midbrain, pons and medulla (Figure 5). Spongiform change is not usually evident in the cerebellum and spinal cord. There is also neuronal degeneration and astrocytosis [39]. Anti-PrP antibodies revealed many different PrP^Sc^ types: plaque-like, perineuronal and subependymal deposits as well as diffuse staining (fine granular) in the cerebellum and the hippocampal formation (Figure 6) [89].

TME PrP^Sc^ deposits have also been described in the spleen, intestine, the mesenteric and retropharyngeal lymph nodes, thymus, rectoanal mucosa-lymphoid tissues, kidney, liver and salivary glands of experimentally infected mink [38,39] (Figure 5).

Two distinct strains, “hyper (HY)” and “drowsy (DY)”, were distinguished after transmission of TME in hamsters characterized by hyper-excitability (HY TME strain) and tremor of the head and shoulders, while DY (drowsy) TME strain caused progressive lethargy and drowsiness (revised by [89]).

### 3.3. Chronic Wasting Disease (CWD)

CWD is a TSE naturally affecting farmed and free-ranging cervids such as mule deer (*Odocoileus hemionus*), white-tailed deer (*Odocoileus virginianus*), Rocky Mountain elk (*Cervus canadensis*), moose (*Alces alces*) and red deer (*Cervus elaphus*); it was first identified in North America and South Korea but reported in Scandinavia in 2016 in wild reindeer (*Rangifer tarandus*), a species never previously found to be naturally infected [94].

The *PRNP* gene is remarkably conserved within the family Cervidae; only 16 amino acid polymorphisms have been reported within the 256 amino acid open reading frame in the third exon of the *PRNP* gene. Although all *PRNP* genotypes can be affected with CWD, some polymorphisms appear to result in longer incubation periods in some species. The polymorphisms Q95H and G96S are related to the reduction of the risk of infection [25]. S96S or G96S and Q95H seem to produce a reduced susceptibility, with a longer survivor period (Table 1) [24], being underrepresented in CWD affected populations. It has been observed that PrP^Sc^ deposition in the brain and other organs progress at a slower rate in deer expressing polymorphisms associated with a lower frequency of CWD natural cases (revised by [95]). In natural CWD, different *PRNP* genotypes showed similar PrP^Sc^ deposition patterns in the brain. Yet the potential effect of the genotype on neuropathology and PrP^Sc^ deposition has been difficult to evaluate in natural disease due to the prion strain variability, routes of exposure and incubation periods. Experimentally, this effect was analyzed showing that in certain brain areas (cerebellum and frontal cortex) different PrP^Sc^ immunolabelling types can occur between different *PRNP* genotypes in white- tailed deer (Q95G96/ Q95G96 (wt/wt), S96/wt, H95/wt and H95/S96) [95].

The histopathologic lesions in the brain of cervids with CWD are similar to those observed in other TSE of animals and humans, characterized by neuronal perikaryon vacuolation, micro cavitation of grey matter, astrogliosis, neuronal degeneration and loss, and PrP positively labelled prion deposits and plaques [96]. While brain lesions occur relatively late in the disease, PrP^Sc^ can be detected earlier than histopathologic vacuolation, and before the animal displays signs of disease, and is therefore a recommendable diagnostic tool [94].

The earliest lesions can be found in the dorsal motor nucleus of the vagus nerve with a mild hypertrophy and proliferation of astrocytes followed by spongiform degeneration of the neuropil and neurons [97]. Subsequently affected areas include the hypothalamic nuclei, followed by adjacent nuclei of the brainstem, thalamus, and olfactory cortex (Figure 5). The basal nuclei, cerebrum, and spinal cord are affected next, while lesions in the hippocampus are minimal. The cerebellum is the last region of the brain to be affected. Amyloid plaques, with pale eosinophilic fibrillar foci within the neuropil, can be seen within brain and spinal cord tissue sections. These plaques can be surrounded by vacuoles on haematoxylin and eosin-stained slides [94]. In the brain, PrP^Sc^ is predominantly found in the grey matter, especially perineuronal and perivascular, in the medulla oblongata, diencephalon, and olfactory cortex. In the medulla oblongata, several neuronal nuclei are affected, but most prominently the dorsal motor nucleus of the vagus nerve. The cerebral cortex and the hippocampus are less heavily affected. Large PrP^Sc^ aggregates in the form of plaque-type are particularly prominent in white-tailed deer [98] **(**Figure 6). 

Pathology and PrP^Sc^ distribution in CWD appear to be very similar to that observed in sheep affected with CS, with the involvement of the lymphoid tissue preceding that of the CNS (Figure 5) [97,99,100,101]. However, CWD pathogenesis seems to differ between deer and elk, with less PrP^Sc^ deposition in the lymphoid tissues of elk compared to deer [94]. Elk have more severe lesions in the thalamus and some white matter areas. Congo red birefringent and PAS-positive amyloid plaques have been seen in deer brain but not in elk [97]. The cerebral cortex and basal nuclei of the elk with CWD show minimal fine spongiform degeneration and astrogliosis with focal distribution. The spongiform degeneration with astrogliosis is more prominent in the thalamus where it forms clusters of coarse vacuoles. Fine spongiosis, often in small clusters, is present in the molecular layer of the cerebellum, in the dorsal nuclei of the pons, and the substantia gelatinosa of the spinal cord. Occasional large neurons in various nuclei of the pons show vacuoles [102]. Amyloid plaques are relatively common and can be detected on haematoxylin and eosin (HE)-stained brain sections, most prominently and with decreasing frequency, in white-tailed deer, mule deer, and elk [103]. Neuronal loss and astrogliosis are minimal except for the molecular layer of the cerebellum, which shows rarefaction of granule cells with no indication of apoptosis [102]. The PrP^Sc^ immunostaining is consistently present in the cerebral cortex, basal nuclei and thalamus. In the cerebellum, the immunostaining is present in both molecular and granule cell layers as well as in the dentate nucleus. In the pons, it is widespread over grey structures, whereas in the spinal cord, it is generally confined to the dorsal part of the dorsal horns [102]. Types of PrP^Sc^ deposition in CWD-affected cervid brains include perineuronal and perivascular accumulation, extracellular plaque-like and granular deposits, and subependymal and subpial deposition [103] (Figure 6).

In lymphoid tissues, PrP^Sc^ is detected in lymph nodes, rectoanal mucosa-associated lymphoid tissue, tonsils, and spleen, particularly in germinal centres, with increased accumulation and throughout for disease [101,104,105,106,107,108] (Figure 5). In addition, PrP^Sc^ and/or infectivity has been demonstrated in blood, saliva, faecal material, and urine, suggesting their important role in the dissemination of the disease [94]. 

#### Novel CWD (Nor16CWD)

Recently, different naturally occurring cases of CWD were observed in 3 moose (*Alces alces*) in Norway (designated atypical CWD, novel CWD or Nor16CWD), that showed molecular and IHC phenotypes differing from those previously described for classical CWD in North America, as well as from previous cases in reindeer in Norway [109]. Immunohistochemistry revealed that the moose shared the same neuropathologic phenotype, characterized by mostly intraneuronal deposition of PrP^Sc^ (Figure 6). This type differed from that observed in reindeer and has not been previously reported in CWD-infected cervids. In the moose, after staining with F99/97.6 and L42, PrP^Sc^ was almost exclusively observed as intraneuronal aggregates, although intra-astrocytic type (multiple small granules scattered in the cytoplasm of astrocyte resembling cells) and intra-microglial type (1 single or a few large granules close to microglia-like nuclei) were also observed in the cerebral cortices and olfactory bulb (Figure 5 and Figure 6). The dorsal motor of the vagus nerve was not remarkably stained, as observed in reindeer. A diffuse or discrete punctate staining was observed in the granular layer of the cerebellum of one moose, with stronger staining in some Golgi neurons. In all 3 moose, the cortical regions showed laminar staining of neurons in all the cell layers, especially in fusiform-shaped neurons. The neurons of the olfactory tubercle from the 3 moose also stained strongly, and some glia associated staining could be observed [109]. No PrP^Sc^ was detected in lymphoid tissue [109] (Figure 5) similar to that described in heterozygote ARR sheep affected with classical scrapie as well as atypical scrapie. In those cases, lymphoid tissues infectivity should be further studied as infectivity of lymphoid tissues was demonstrated even with no detectable PrP^Sc^ in atypical scrapie [84].

Moreover, Western blot revealed a PrP^Sc^ electrophoretic pattern distinguishable from previous CWD cases and known ruminant prion diseases in Europe, with the possible exception of sheep CH1641. These findings suggest that these cases in moose represent a novel type of CWD. Transmission studies in several rodent models are ongoing to clarify if this different phenotype could reflect the presence of a new cervid prion strain in moose from Norway) [109].

### 3.4. Bovine Spongiform Encephalopathy (BSE)

#### 3.4.1. Cattle

BSE was first recognized in cattle in Great Britain in 1986 [110] and soon spread to at least 28 other countries, mostly in Europe. Although the origin of BSE remains uncertain it is widely believed that cattle feed prepared from prion-infected animal tissues was the source of its dissemination in cattle populations [111,112].

BSE was recognized as a zoonotic disease posing a serious risk to human and animal health as it was discovered a pathogenic relationship between BSE and a new fatal neurodegenerative disorder in humans, called variant Creutzfeldt-Jakob disease (vCJD) [113,114]. In addition to this classical form of BSE (C-BSE), two atypical BSE prions (L-BSE and H-BSE) were identified in France and Italy in 2004 categorized based on low and high apparent molecular masses of unglycosylated protease-resistant PrP (pathogenic prion protein) on Western blots. L-type BSE is also known as bovine amyloidotic spongiform encephalopathy (BASE) due to the presence of abundant amyloid plaques [115,116]. These atypical forms are rare and believed to occur spontaneously in older cattle populations. Between 2001–2019, a total of 64 H-BSE and 69 L-BSE were reported in European Union (EU) and non-EU reporting countries in comparison to C-BSE that totalizes 190,469 cases since its identification up to 2019 being the United Kingdom, Ireland and Portugal the most affected countries with this classical form of the disease [117].

Regarding genetic susceptibility and its influence on BSE pathology, there are fewer polymorphisms in the bovine *PRNP* gene than in other TSE affected species. Only three amongst these variants, a 23 bp indel in the *PRNP* promoter) [27], a 12 bp indel in the first intron [28] and an E211K polymorphism [29] are known to be putatively linked to susceptibility to BSE [118] (Table 1) but not to variability in spongiform changes and PrP^Sc^ as far as it is known.

Although there is a consistent vacuolar pattern in the brains of BSE-affected animals, bioassay data support the hypothesis that C-BSE and its atypical forms are biologically distinct strains presenting clinical differences in the evolution of the disease as well as different histopathologic features [119,120,121]. 

The basis for confirmation of the diagnosis of clinical C-BSE is its characteristic spongiform changes that comprise mainly neuropil and neuronal vacuolation in certain anatomic nuclei of the medulla oblongata at the level of the obex [122,123]. 

Vacuoles are singular or multiple, round or ovoid and without content in the neuropil of grey matter and within neuronal perikarya. Both neuropil vacuolation and neuronal perikaryonal vacuolation are bilaterally distributed and usually symmetrical with a consistent pattern of severity relative to distribution throughout the brain [122,124]. Both forms of neuroparenchymal vacuolation reach their greatest intensity in specific anatomic nuclei of the medulla oblongata at the level of the obex, predominantly in the spinal tract nucleus of the trigeminal nerve (NSTV) and in the tract of the solitary nucleus (NST). Vacuolation in the pons is characterized by neuronal vacuolation mainly in neurons of the vestibular nuclei. In the midbrain, vacuolation is more frequent in the neuropil of central grey matter (reviewed by [125]) (Figure 5). Intraneuronal vacuolation alone, in the absence of neuropil vacuolation, is not BSE confirmatory since vacuolated neurons, particularly in certain locations, such as the red nucleus, may be an incidental finding in cattle [126].

A strong astrocytic response, assessed by immunohistochemistry for glial acidic fibrillary protein, was observed at all levels, particularly in areas with vacuolation [122].

The most frequent PrP^Sc^ immunolabelling types scattered throughout the encephalon are ([122]; reviewed by [125]) (Figure 5 and Figure 6):Intraneuronal—often observed in the dorsal motor nucleus of the vagus nerve (DMV), reticular formation, olivary nuclei, vestibular complex, pontine and thalamic nuclei, and hypothalamus;Intraglial;Stellate—predominantly present in the central grey matter, the cerebral lamina, and within the medial pontine nuclei in the cerebral cortex, thalamus, and obex;Fine granular—seen in the neuropil of the DMV, NST and the thalamic nuclei;Aggregates;Linear—noticed particularly at the level of the reticular formation of the brainstem;Perineuronal—detected in the caudate and putamen nuclei of the basal ganglia and in the DMV;

As described by Wells and colleagues [122], no PrP^Sc^ deposition was observed within the neurons in the DMV [127]. Amyloid and plaques-type are extremely rare in cattle brains. PrP^Sc^ accumulation generally correlates in distribution with vacuolation, although in individual cow brains the former is usually more widely distributed than vacuolation.

In cattle, PrP^Sc^ is mostly confined to the nervous system (CNS) (Figure 5), although limited involvement of the Peyer’s patches in the distal portion of the ileum has been documented in experimentally induced and naturally acquired cases of BSE as well as in skeletal muscle [121].

When the feed-borne source was established as the better hypothesis for BSE emergence [128], several efforts were taken to guaranty that the total feed ban has been effectively enforced. It is widely accepted that the initial feed bans, while very effective at reducing numbers of cases, were not as robust as they needed to be, and further reinforced bans were implemented.

Nevertheless, a total of 61 C-BSE cases emerged on animals born after the reinforced feed ban (BARB) (1 January 2001) [117]. If the total feed ban has been effectively enforced, these BARB cases should differ in some way from the cases born before the total feed ban. However, it has been reported that the clinical signs, tissue histopathology, and prion protein immunohistochemistry of BARBs cases are identical to those observed in all other bovine C-BSE cases in the UK, indicating that it is the same disease as before the reinforced feed ban was introduced (reviewed by [129]). 

#### Atypical BSE (L-BSE and H-BSE)

Little is known about features of naturally occurring atypical BSE; thus, most of the data are from experimentally infected animal. Postmortem findings of cattle inoculated with H-type BSE and L-type BSE by the intracerebral route show vacuolar lesions consistent with TSE throughout the neuraxis in both the H-type and L-type BSE cases including the neuropil vacuolation at the level of the obex (NST) like in C-BSE [130] (Figure 5).

However, in natural cases of L-BSE (BASE), spongiosis is not consistently found in the brainstem, at the level of the obex or in more rostral areas. The frontal, parietal and occipital cortices are apparently spared, and no vacuolation is detected in the olfactory bulb, piriform cortex, and hippocampus [116]. More severe involvement of the central grey matter (periaqueductal grey) and rostral colliculus, but not the vestibular nuclear complex, has been observed in experimental cases of L-BSE (Figure 5). Additional brain areas, including the olfactory area, amygdala, hippocampus, and dorsal horns of the spinal cord, are severely involved. The ventral and dorsal roots do not show major pathological changes [120].

Vacuolation is generally observed in all the brain areas in experimental H-type BSE, namely in the thalamic nuclei and the neuropil of the central grey matter of the midbrain; however, mild vacuolation can also be present in the caudal cerebral and cerebellar cortices (Figure 5). Spongy changes in the vestibular and pontine nuclei are not as prominent as those seen in the other brainstem nuclei [121].

At the level of the obex, there are also no obvious differences in the amount and distribution of PrP^Sc^ immunolabelling between atypical BSE and C-type BSE, nevertheless, there are subtle phenotype-specific features in this area too, specifically labelling in white matter tracts in the H-type BSE cases, and small aggregates of immunolabelling throughout the reticular formation in L-type BSE cases [130].

In the other areas of the neuroaxis, there is a widespread PrP^Sc^ immunolabelling in both the H- and L-type BSE cases, but with distinct types from one another and distinguishable from that seen in C-type BSE [85,131]. The L-BSE shows extensive small plaque-like PrP^Sc^ deposits, its distinctive feature, predominantly located in the thalamus, subcortical white matter, the deeper layers of the cerebral cortexes and the olfactory bulb. These PrP^Sc^ positive amyloid plaques are dense, unicentric, or less frequently multicentric, round structures up to 25 μm in diameter with a pale core and a dark radial periphery. Other prevalent PrP^Sc^ types are the granular, mildly present in the hypoglossal and olivary nucleus and moderately observed at the level of DMV nucleus, NST, NSTV, and reticular formation, along with the stellate, intraneuronal, perineuronal and linear in different brain areas in BASE cases (reviewed by [125]) (Figure 6).

In natural H-type BSE fine granular, intraneuronal, linear, intraglial, and punctate PrP^Sc^ deposits in the brainstem are the most characteristic types mainly detected at the level of the DMV, NST, NSTV and in the reticular formation, yet with a certain variability in PrP^Sc^ distribution [132]. In experimental H-type BSE, large amounts of PrP^Sc^ are diffusely deposited throughout the CNS, including the cerebral cortex, basal nuclei, thalamus, hypothalamus, brainstem and spinal cord. Fine granular and aggregates type’s deposits in the neuropil of the grey matter throughout the brain and spinal cord are the most conspicuous type of PrP^Sc^ deposition. However, linear, perineuronal, and intraneuronal types of PrP^Sc^ can be observed in the cerebral cortex, basal nuclei, thalamus, and brainstem. Stellate type PrP^Sc^ deposition is predominantly identified in the cerebral cortex, basal nuclei, thalamus, hypothalamus and hippocampus, and often in the cerebellar cortex, but it is not visible in the brainstem and spinal cord [121]. Intraglial-type PrP^Sc^ deposition is highly consistent throughout the white matter tracts of the brainstem [132,133,134] (Figure 6). Some animals present PrP^Sc^ -positive plaques scattered throughout the cerebral white matter [121].

L-type BSE cases show immunolabelling in the cerebellum, in both the molecular and granular layers, similar to that described for atypical scrapie in sheep [130,132,134]. In H-type BSE the immunolabelling in the molecular and granular layers of the cerebellum is minimal and less uniformly distributed compared to L-type BSE, but there is a remarkable widespread glial labelling throughout the white matter of the spinal cord and the cerebellum [130]. Thus, the cerebellum is an important neuroanatomical target region to discriminate atypical BSE forms. Widespread immunolabelling was seen throughout the dorsal and ventral horns at all levels of the spinal cord as in C-type BSE. Immunolabelling was also present in the muscle spindles of the extraocular muscles and the trigeminal ganglion of both types. Where muscle spindles were found in other muscles (triceps in one H-type BSE case, and medial gluteal in one L-type BSE case) these were also immunolabelled [130]. No immunolabelling was observed in the lymphoid tissues or the enteric nervous system [130].

#### 3.4.2. Zoo Ruminants and Non-Human Primates

During the BSE epidemic in cattle, several exotic zoo ruminants, greater kudu (*Tragelaphus strepsiceros*), eland (*Taurotragus oryx*), nyala (*Tragelaphus angasi*), gemsbok (*Oryx gazella*), Arabian oryx (*Oryx leucoryx*), a scimitar-horned oryx (*Oryx dammah*) and a bison (*Bison bison*), succumbed to a spongiform encephalopathy (reviewed by [39]).

Although the detailed distribution of neuropathological changes in the exotic ruminants has not been always reported, there were clear differences in the severity of involvement of certain anatomic nuclei compared to C-type BSE [124]. In the nyala case, the vacuolization of the DMV was more intense than observed in BSE [124]. Of all the species exposed naturally to the BSE agent, the greater kudu appears to be the most susceptible to the disease and the results of mouse bioassay studies show that, contrary to findings in cattle with BSE in which the tissue distribution of infectivity is the most limited recorded for any of TSE, infectivity in greater kudu with BSE is distributed in a wide range of tissues [135].

The geographic and temporal association with BSE suggested possible links to the epidemic as zoo species were exposed to BSE-contaminated meat and bone meal [39]. Most affected animals had consumed diets that included ruminant-derived meat and bone meal.

This hypothesis was confirmed by the transmission assays of mice inoculated with brain homogenates from TSE-infected kudu and nyala that showed incubation period, lesion profile and biochemical PrP^Sc^ electrophoretic profile similar to that seen in mice inoculated with BSE [136].

Among the primates, several deaths were recorded in zoo lemurs and monkey rhesus between 1989–1998 in France due to neurological illnesses associated with spongiform encephalopathy and the presence of PrP^Sc^ (reviewed by [137]). The similar neuropathology and distribution of PrP^Sc^ in orally infected experimental lemurs and spontaneously affected zoo lemurs, together with the epidemiological observations confirmed that the occurrence of spongiform encephalopathy in those animals was related to a diet supplemented with meat protein that until 1996 had very likely included rendered BSE contaminated meat and bone meal [137].

Concerning the neuropathological features of the monkey rhesus, spongiform changes were prominent in many grey structures, including the cerebral cortex, striatum, thalamus, and cerebellar cortex. Changes in the white matter were minor and were related to enlargement of nerve-cell processes without demyelination. The spongiform changes were associated with PrP^Sc^ and β-protein deposition. These pathological changes recorded in the deep layers of the cerebral cortex do not fit with the BSE-infected primates that showed the most intense lesions in the thalamus and striatum and numerous characteristically florid PrP^Sc^ [138].

#### 3.4.3. Small Ruminants

The spread of the BSE agent in small ruminants has been considered a major threat in recent years because sheep and goats were exposed to the same contaminated feedstuffs as cattle during the BSE epidemic before the exclusion of meat and bone meal from ruminant feedstuffs.

To date, there are no reports of naturally occurring C-type BSE in sheep, whereas two cases were confirmed in goats in France and the UK [139,140]. According to the data from these studies, the PrP^Sc^ distribution was limited to the brain in these natural cases, but experimental studies to better characterize the C-type BSE phenotype in the event that it is transmitted to sheep and goats have shown that both species are readily infected with the agent of C-type BSE and that lesion profile and PrP^Sc^ distribution seems to be uniform, genotype independent and very similar to that observed in classical scrapie [141,142], including the PrP^Sc^ deposition in lymphoid tissue [143,144].

Histopathology of BSE in orally infected sheep showed vacuolation throughout the brainstem, including the raphe, DVM, olivary and facial nuclei, thalamus and hypothalamus, but there was no evidence of cortical lesions (Figure 5). The BSE orally dosed goats had vacuolation in the midbrain and thalamus. Very little vacuolation was identified in cortical areas [141].

Immunostaining for PrP^Sc^ was prominent throughout the brain in BSE infected sheep and goats. The thalamic nuclei, especially the ventral and the hypothalamus showed intense immunostaining, as did many nuclei in the medulla, for example, the dorsal vagus. The basal ganglia were also heavily immunostained, but in the cortex, only the inner nuclear layers were strongly stained [141]. Experimental ovine BSE is characterized by striking immunolabelling of PrP^Sc^, with prominent granular intraneuronal accumulations and distinctive extracellular labelling in the form of a conspicuous perineuronal type found in the striatum and marked linear labelling pattern in both the striatum and the substantia nigra [142] (Figure 6). These features were all present in goat BSE, which was therefore essentially indistinguishable from sheep BSE [145].

The cellular and neuroanatomical distribution of PrP^Sc^ differed in BSE- and scrapie- infected sheep and goat. The intra-neuronal PrP^Sc^ in both goat and sheep BSE was labelled only by antibodies recognizing epitopes located C-terminally of residue His99 [145], due to the variation in the N-terminal truncation site of the PrP^Sc^ occurring within different populations of cells [144].

### 3.5. Feline Spongiform Encephalopathy (FSE)

Domestic cats and captive wild felids, like cattle, are susceptible to prion disease which, in these species, is designated Feline Spongiform Encephalopathy (FSE). The first report of naturally occurring FSE was published in 1990 in the United Kingdom (reviewed by [146]). Since then, cases were diagnosed in several countries affecting domestic cats as well as captive wild felids including cheetahs, lions, tigers and pumas. According to Bruce and colleagues [136], strain typing studies carried out in mice and biochemical profile assays provided strong evidence of an association between these cases and BSE. These results suggest that felids were infected through ingestion of BSE contaminated food.

Similarly to other prion diseases, FSE is an insidious and progressive disease clinically characterized by neurological signs such as behavioural changes, ataxia, abnormal gait (circling, head tilt), blindness, polydipsia and hypersalivation [147]. Since there are no macroscopic lesions, FSE diagnosis requires histological examination as well as immunohistochemical and biochemical profile studies.

In cats, moderate to severe widespread vacuolation of neuropil is seen namely in corpus geniculatum medialis, and the basal nuclei but also in gyrus dentatus of the hippocampus, thalamus, deep layers of cerebral and cerebellar cortexes and brainstem. [148,149]. Vacuolation is also present in neuronal perikarya more caudally, namely in raphe nucleus, red nucleus, vestibular complex, reticular formation, dorsal vagal nucleus and ventral horn of spinal cord particularly in the dorsal vagal nucleus [148,150] (Figure 5). There is vacuolation in the white matter mainly in the medulla [148].

Immunohistochemical detection of PrP^Sc^ showed a strong and widespread positivity throughout the encephalon as granular, aggregates, linear [148] and plaque-like type as well as in association with some neurons (Figure 1 and Figure 6) [150].

Heavy PrP^Sc^ immunostaining is also detected in some extra brain tissues: retina, optic nerve, pars nervosa of the pituitary gland, adrenal medulla, trigeminal ganglia and myenteric plexus of the small intestine [149].

Both the histological lesions and the PrP^Sc^ distribution seen in FSE-affected cats are similar to that described in puma and cheetahs, but in these latter species PrP^Sc^ accumulation was detected in the kidney and lymph nodes in contrast to FSE affected cats [149,151,152] (Figure 5).

### 3.6. Prion Diseases in Dromedary Camels (CPD)

Dromedary camel or Arabian camel (*Camelus dromedarius*) is one of the three surviving species of camel and represents 94% of the world camel population. They are present in Northern and Eastern Africa, the Middle East, part of Asia and also in Australia. In the latter country, the presence of this species is related to animals originally imported from British India and Afghanistan for use in transport and construction, since mid-19th century [153].

The Bactrian camel (*Camelus bactrianus*), distributed mainly in Central Asia, and the Wild Bactrian camel (*Camelus ferus*), distributed in Northwest China and Mongolia, are the remaining two species of the family Camelidae [153]. They are of extreme importance due to their milk and meat, constituting an excellent food resource in arid and semi-arid climates [154].

Recently, a prion disease, named Camel Prion Disease (CPD), was first identified in dromedary camels in Algeria in 2018 [32] and then in Tunisia, in the Tataouine region, in 2019 [155]. The first dromedary camel cases, from a Saharian population in Ouargla (Southeastern Algeria) were identified in a routine antemortem inspection when brought for slaughter at the Ouargla abattoir, one of the largest in Algeria [32].

The animals arrived at the abattoir showing weight loss, behavioural abnormalities (observed in the early stages of disease) and also neurologic signs, such as tremors, aggressiveness, hyperreactivity, typical down and upward movements of the head, uncertain gait, ataxia of the hind limbs, falling and difficulty in rising from a lying position. According to local breeders, the disease could extent from 3 to 8 months and its thought to be present since the 1980s [32]. The affected animals had the same *PRNP* genotype showing 100% nt identity with the *PRNP* sequence already reported for dromedary camels [32].

Histopathology showed spongiform change, gliosis and neuronal loss in symptomatic animals but not in asymptomatic ones. Vacuolation was always seen in the neuropil but it could also be found in neuronal bodies. Confluent vacuoles were rarely observed. Neurodegenerative changes were consistent in the grey matter of subcortical brain areas (striatum, thalamus, midbrain, and pons) while rare in the white matter (Figure 5). The cortical brain areas and the cerebellum were variably involved presenting vacuolation in cingulate, piriform, and frontal cortices and only in the molecular layer of the cerebellum. The cervical medulla showed no spongiform changes [32].

In medulla oblongata, moderate vacuolation was observed, particularly in the vestibular and the olivary nucleus; the nucleus of the solitary tract and the hypoglossal nucleus were less often affected [32].

Immunohistochemical staining showed PrP^Sc^ deposition associated with vacuolation but also in areas less or not affected by spongiosis such as the nucleus of the solitary tract, the hypoglossal nucleus, the pyramidal cells of the hippocampus, the granular layer of the cerebellum, the Purkinje cells and several white matter areas.

The most frequent detected PrP^Sc^ types were intraneuronal, intraglial, synaptic/punctuate (equivalent to fine granular), perineuronal, linear, and perivascular (Figure 6). In pons and medulla oblongata, an atypical intracellular pattern was observed in which PrP^Sc^ filled the whole cytoplasm. PrP^Sc^ was absent in the brain of the asymptomatic dromedary [32].

PrP^Sc^ was present in all lymph nodes collected from one animal, suggesting extra-neuronal pathogenesis and so, a potential excretion that may result in transmission between animals [32] (Figure 5).

The western blot characterization has shown that dromedary camels’ PrP^Sc^ is less glycosylated than those of CS. It presents a monoglycosylated dominant PrP^Sc^ and an apparent molecular weight slightly higher than CS and clearly higher than BSE and sheep passaged BSE [32].

The origin of CPD is still unknown but it may be associated with the exportation of meat and bone meal from BSE affected countries and subsequent contamination of animal feed. However, dromedaries are usually not fed commercial feed. On the other hand, these animals are frequently grazed with sheep and goats; hence, CPD’s origin could be related to scrapie. However, there is no scrapie surveillance program in Algeria and no cases were reported so far. To clarify these questions, bioassays are being performed in rodent models for an exhaustive strain characterization [32].

Upon detection of this new disease, the OIE acted to study its impact and to decide if it would be considered an emerging disease. For that, two ad hoc groups were consulted: one for the evaluation of BSE risk status and the other about camelids. As there are still few data concerning this disease it was not possible to conclude about its impact on animal and public health. However, surveillance in countries with affected or not dromedary’s population is crucial for collecting information needed for risk assessment.

Two projects are ongoing for the coordinated surveillance of CPD. One launched by the CAMENET (Camel Middle East Network) and the other by the EFRAN (Enhancing Research for African Network) [155].

## 4. Neuroinflammation

Along with neurodegeneration, chronic neuroinflammation is a hallmark of prion brain diseases, but the process that leads to it is not yet fully understood. Recent research shows that during the accumulation of misfolded PrP^Sc^ both anti- and pro-inflammatory factors and molecules are active in the prion brain [156], reaching a stage of chronic inflammation that is likely to contribute to prion pathogenesis [5]. It appears that the vast majority of these factors is produced by cells within the CNS, namely microglial cells, cells that also exhibit PrP^Sc^ deposition [66], as it was demonstrated that the arrival of leucocytes from the periphery is limited and only detectable at the later stages of clinical disease [35,157,158]. As recorded for other diseases, microglia are the brain guardians, with several subpopulations, multifaceted, with neuroprotective but also neurotoxic properties [5], displaying region-dependent homeostatic transcriptional identities, observed during prion-associated neuroinflammation [159]. As such, the role of microglia in this group of diseases remains undefined, as they could be an actor in the pathogenesis or a simple consequence of the disease [160]. Besides, in scrapie, several chemokines were identified since the early asymptomatic stage and probably were enrolled in the disease progression. The inflammasome activation by the CD36 molecules also promotes the activation of several proinflammatory cytokines. All these factors (chemokines, inflammasome and proinflammatory cytokines) are crucial in the microglia recruitment and the inflammation and neuronal damage promoted by the PrP^Sc^ deposition [160]. Due to its nature and course, it is very difficult to address the neuroinflammation component of the disease in natural occurring prion cases. Thus, information regarding this aspect of the disease relies mainly on experimental studies that provide valuable and challenging data about the process.

During prion infection, and before reaching CNS, prions are first detected in lymphoid tissues, frequently associated with follicular dendritic cells (FDCs). They then progress through the nerves of the autonomic nervous system and finally reach the CNS where no apparent peripheral immune response is registered, albeit the activation of microglia and astrocytes [161]. In some experimental studies, the observation of increasing density of activated microglia cells, namely throughout the cerebellar cortex, as observed in natural cases, is associated with the upregulation of the TNF-α Receptor type-1 [162]. Single-cell transcriptional profiling screening 1/2 million cells revealed seven molecularly distinct and regionally restricted astrocyte types [163]. Astrocytes are the cells directly implicated in the direct neurotoxic effects related to the human and animal prions, namely the scrapie, and act as inflammation promoters [162]. In fact, astrocyte function pathway is activated in prion infection prior to the activated microglia or neuron and neurotransmission pathways [159].

Microgliosis is observed in infected brains even before the neuronal loss and spongiform change [5], microglial cells phagocytize prions and promote apoptotic cell clearance of neurons, a process mediated by the secretion of milk fat globule epidermal growth factor 8 (MFGE8) by astrocytes [164]. However, in vivo, this protective role of microglia becomes insufficient, which can induce microglia to convert from the phagocytic M2 phenotype into the pro-inflammatory M1 phenotype [165]. This altered phenotype may exacerbate the secretion of cytotoxic mediators and contributes to the spreading of prions, while increases the secretion of pro-inflammatory mediators by microglial cells [161].

Several studies allowed the identification of the inflammatory profile regarding the transcriptome and protein changes occurring in the prion- infected brain, some of them using scrapie strain infected mice [166]. Pro-inflammatory cytokines and chemokines, among which IL-1α and β, IL-12p40, TNF, CCL2–CCL6, and CXCL10, are increased in the brains of mice with the clinical disease [156,167]. High-density qRT-PC studies investigating different times of the disease progression revealed overexpression of inflammatory genes *Il1rn* [IL-1Ra], *Ccl8* [CCL8/MCP-2], *Tnfsf11* [Tnfsf11/RANKL], and *Osm* [OSM]) at the preclinical stage in scrapie strain 22L-infected mice). The upregulation of these genes occurred before clinical signs [168] and continued with disease progression. Later on, *Oas1a*, *Isg15*, *Olr1*, and *Ccl5*, among others, were found to be increased as well. Overall, the transcription of inflammatory mediators increases as the disease evolves to display the different clinical phases and endpoint [5]. The analysis of the transduction pathways in prion infected brain revealed that the AK-STAT and NF-κB pathways are substantially activated during the disease, being responsible for the transcription of many of these genes. Phosphorylated STAT proteins can act independently, but in addition they can also act synergistically with NF-κB [168], thus contributing to the enhancement of the expression of acute phase proteins such as haptoglobin, ceruloplasmin, α1-antichymotrypsin, and serum amyloid A [169,170].

On the other hand, several of the mentioned inflammatory mediators have the potential to induce damage to the CNS. The triggering of apoptosis in cells is controlled by the expression of *Oas1a*, *Isg15*, *Tnfsf1*1, *Olr1*, and *Ccl5* [171,172,173], cytokines like CCL2, CCL7, and CCL8 can attract monocytes [174], and it has also been shown that the expression of Cxcl10, Ccl2, A2m, and Tnf can contribute to neurotoxicity in other disease models [175,176,177,178].

The importance of region-specific astrocyte and microglial identities during prion- associated neuroinflammation was highlighted by recent studies: specific brain regions, as well as region-specific homeostatic identities are preserved during the preclinical stages of the disease. This is related to different prion strains and cell tropism. However, as disease progresses and the clinical signs arise, the region-specific homeostatic transcriptome signatures are replaced by the region-independent neuroinflammation signature, regardless of prion strain and cell tropism [159].These and other studies suggest that the inflammatory component of the disease concurs to neurodegeneration [167], by inducing cellular damage in the CNS and stimulating its surveillance system to display gliosis and activate astroglia and microglia observed in early disease before vacuolar pathology or clinical signs [35]. Ultimately, an inflammatory self-perpetuation cycle is then established in the prion infected brain, in which neuronal damage and astrocyte and microglial activation concur to the pathobiology of the disease and its outcome. The increase microgliosis is closely associated with spongiosis and astrogliosis, and the correct understanding of its functions must be, in the future, related with therapies, including the microglia reprogram throughout the prion clearance mechanisms, instead of neurotoxicity associated with these cell populations in the later stages of prion diseases [5].

## Figures and Tables

**Figure 1 biomolecules-11-00466-f001:**
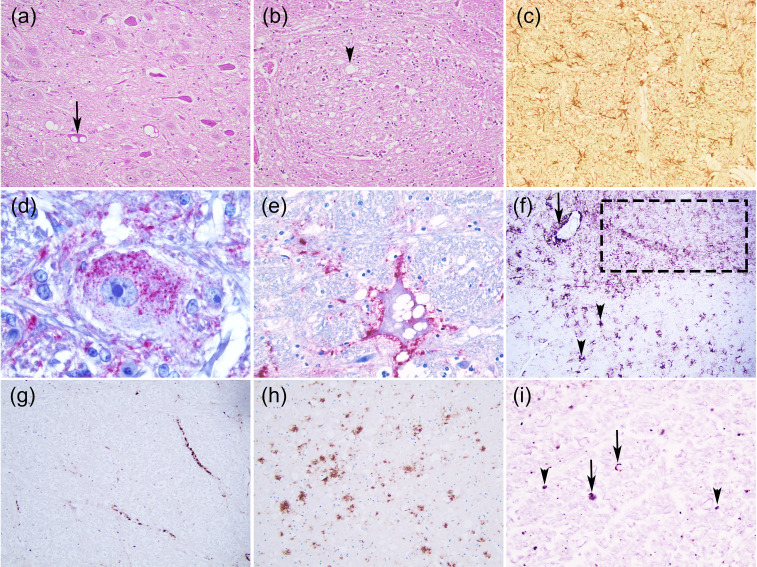
Neurohistopathological features of animal prion diseases. (**a**) Neuronal vacuolation (arrow) (Classical scrapie, sheep, medulla oblongata, dorsal vagal nucleus); (**b**) Neuropil vacuolation (arrow head) (Classical BSE, bovine, medulla oblongata, nucleus of the solitary tract); (**c**) Astrogliosis (Atypical scrapie, sheep, medulla oblongata, GFAP Polyclonal antibody, DAKO, 1:1000 dilution, x200); (**d**–**i**) Some PrP^Sc^ deposition types: (**d**) Intraneuronal (Feline spongiform encephalopathy, cat, medulla oblongata, reticular formation, 3F4 PrP Residues 109–111 Monoclonal antibody, DAKO, 1:300 dilution, x1000); (**e**) Perineuronal (BSE, bovine, medulla oblongata, reticular formation, 6H4 PrP Residues 144–152 Monoclonal antibody, Prionics AG, dilution 1:1000 x400); (**f**) Fine granular (dot line), stellate (arrow head), perivascular (arrow) (Classical scrapie, sheep, medulla oblongata, 2G11 PrP Residues 146-R154-R171-182 Monoclonal antibody, Pourquier Institute, 1:200 dilution, x100); (**g**) Linear (Classical BSE, bovine, medulla oblongata, reticular formation, 12F10 PrP Residues 142–160 Monoclonal antibody, SPBIO, 1:200 dilution, x200); (**h**) Plaque-like (cerebral cortex section from the CWD Proficiency testing 2008 organized by the European Reference Laboratory for TSEs-APHA, Weybridge, 2G11 PrP Monoclonal antibody, 1:200 dilution, x100; (**i**) Punctuate (arrow head) and globular (arrow) (Atypical scrapie, sheep, medula oblongata, pyramidal tract, 2G11 PrP Monoclonal antibody, 1:200 dilution). (**a**–**b**) Haematoxylin and eosin (H & E) stain, x200; (**d**–**e**) *StreptABC-AP, Nova fucsina,* Mayer’s Haematoxylin; (**c**, **f**–**i**) *Vectastain Elite –HRP,* DAB, Mayer’s Haematoxylin.

**Figure 2 biomolecules-11-00466-f002:**
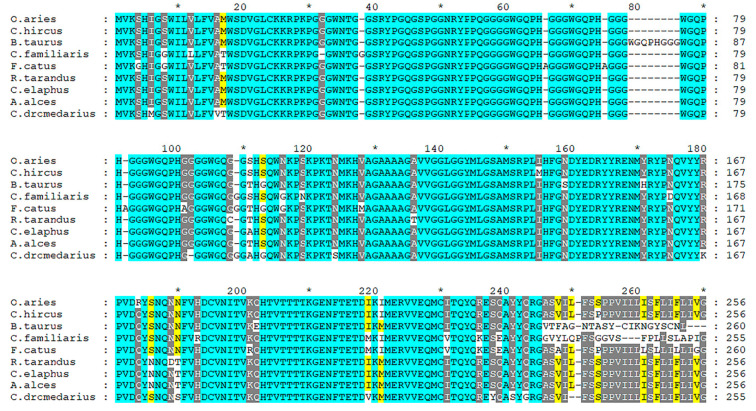
Multiple alignment of PrP protein from *Ovis aries (*ENSOARP00020014736)*, Capra hircus (*CAA63050.1)*, Bos Taurus (*ENSBTAP00000069134)*, Canis familiaris (*ENSCAFP00000009107)*, Felis catus (*ENSFCAP00000030786)*, Rangifer tarandus (*ABS87897.1)*, Cervus elaphus (*AAU93885.1)*, Alces alces* (AZB50215.1) and *Camelus dromedarius* (CAA70901.1). The alignment was performed using T-Coffee program [9] and edited using Genedoc software [10].

**Figure 3 biomolecules-11-00466-f003:**
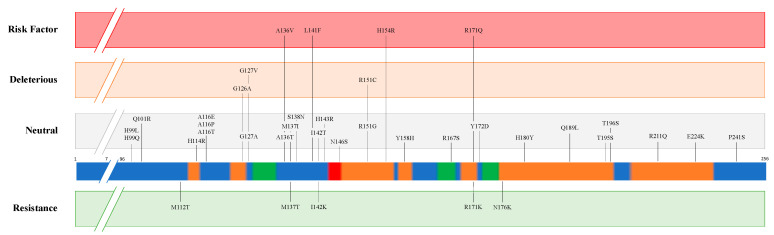
Missense mutations of prion protein (PrP) in *Ovis aries*. The coloured bar represents structural features of the protein according to UniProt (P23907), including α-helix (orange), β-strand (green) and turn (red). Variants are classified above and below the bar as ”Risk factor” and ”Resistance” according to in vivo and in vitro analyses. The other variants were analysed using an in-silico approach through PROVEAN algorithm, and classified as ”Deleterious” and ”Neutral”.

**Figure 4 biomolecules-11-00466-f004:**
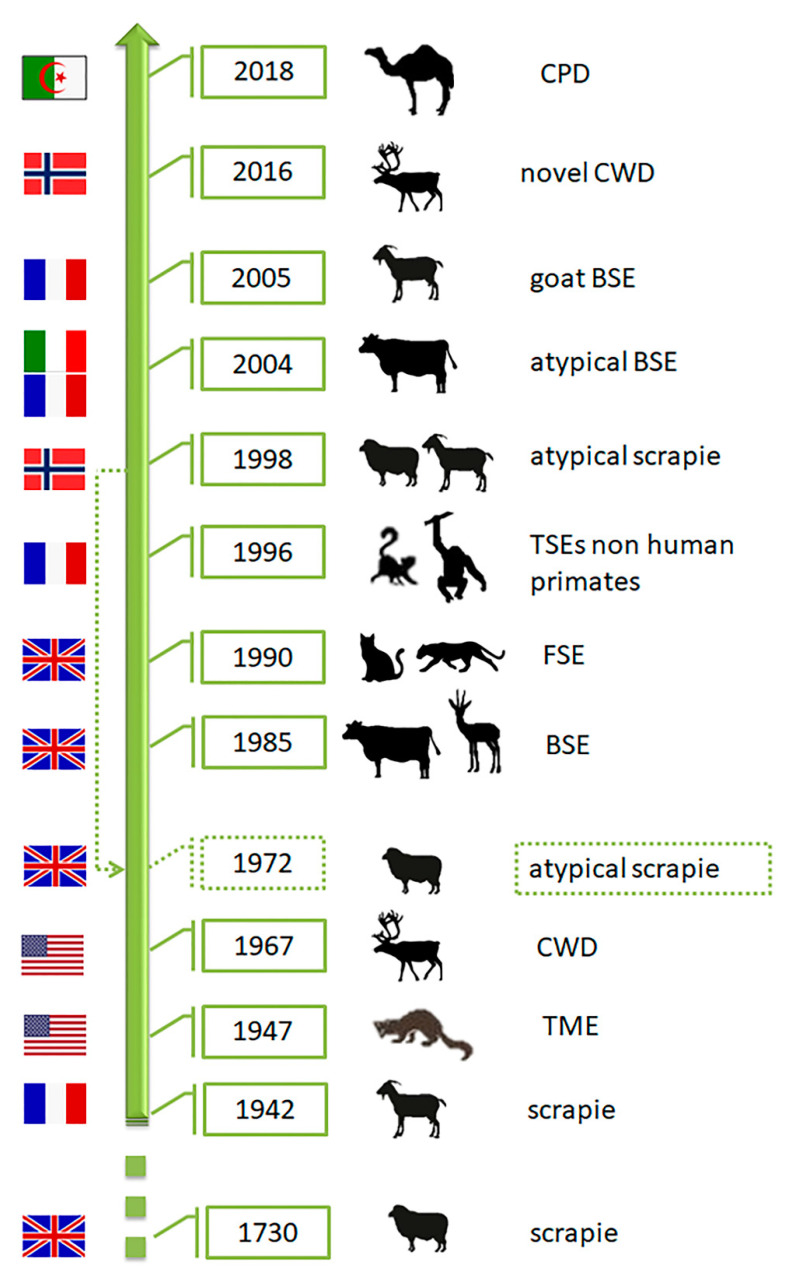
Chronology of identification of animal prion diseases. TME-transmissible mink encephalopathy; CWD—chronic wasting disease; BSE-bovine spongiform encephalopathy; FSE—feline spongiform encephalopathy; CPD—camel prion disease. The flags represent the country of the first report of the disease. The dashed arrow indicates retrospective identification of atypical scrapie in 1972 (Silhouettes from Freepik.com and img.inkfrog.com).

**Figure 5 biomolecules-11-00466-f005:**
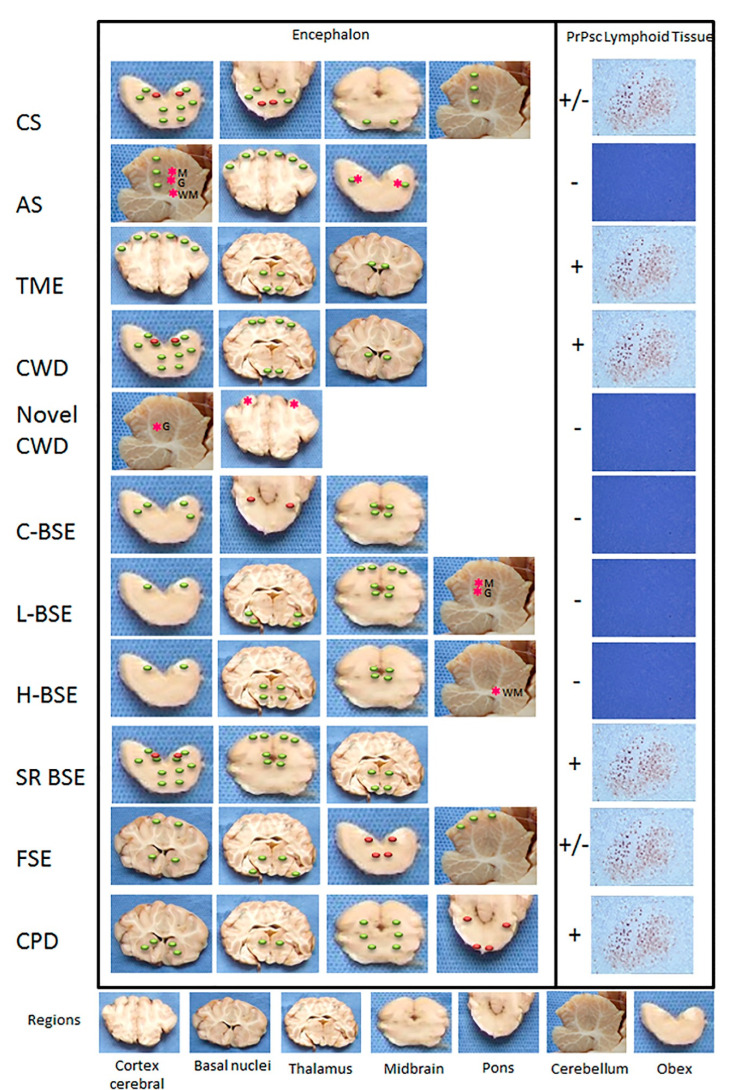
Major distinctive distribution of lesions and PrP^Sc^ in animal prion diseases. CS—classical scrapie; AS—atypical scrapie; TME-transmissible mink encephalopathy; CWD—chronic wasting disease; C-BSE—classical bovine spongiform encephalopathy; L-BSE-atypical Low BSE; H-BSE—atypical High BSE; SR BSE—small ruminant BSE; FSE—feline spongiform encephalopathy; CPD—camel prion disease. M Molecular layer; G Granular layer; WM white matter; +, presence of PrP^Sc^ by immunohistochemistry; +/−, presence of PrP^Sc^ depending on *PRNP* genotype or species; −, absence of detectable PrP^Sc^ by immunohistochemistry; red dots, neuronal vacuolation; green dots, neuropil vacuolation; red asterisk, PrP^Sc^ deposition.

**Figure 6 biomolecules-11-00466-f006:**
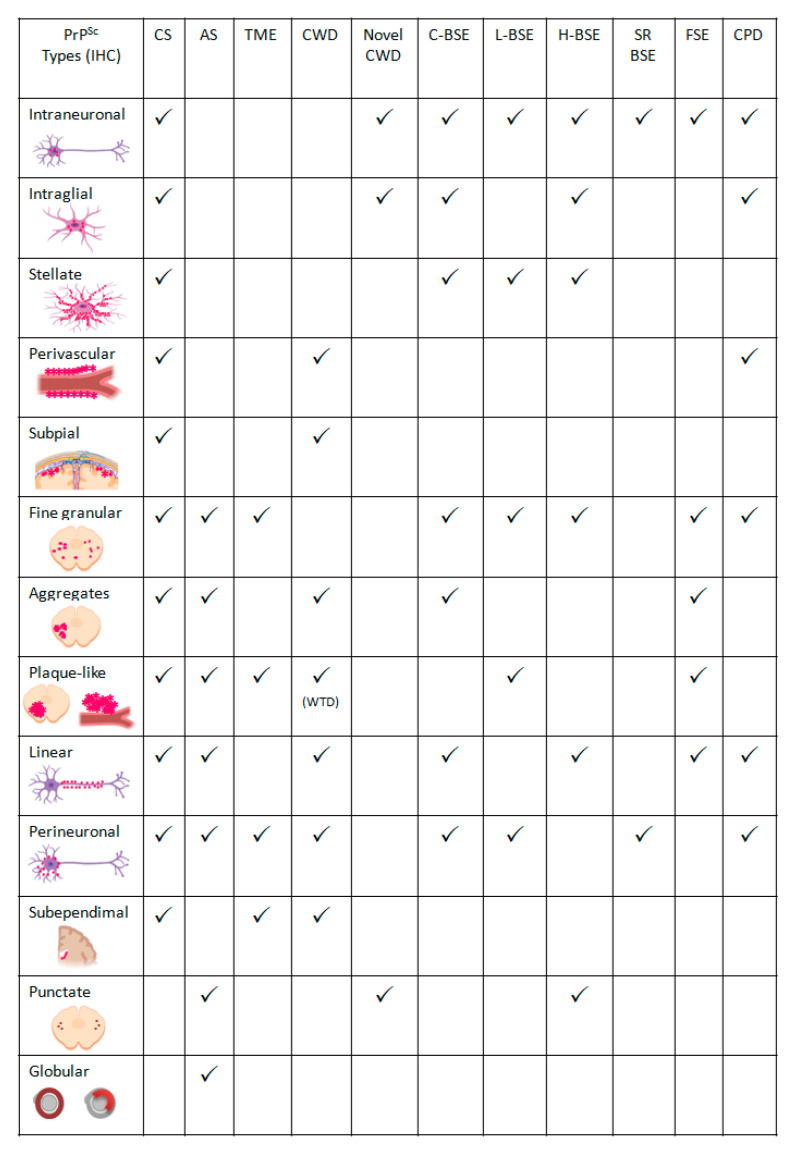
Summary of the most frequent PrP^Sc^ immunolabelling types in animal prion diseases. CS—classical scrapie; AS—atypical scrapie; TME-transmissible mink encephalopathy; CWD—chronic wasting disease; C-BSE—classical bovine spongiform encephalopathy; L-BSE-atypical Low BSE; H-BSE—atypical High BSE; SR BSE—small ruminant BSE; FSE—feline spongiform encephalopathy; CPD—camel prion disease. WTD—white-tailed deer. The figures of PrP^Sc^ types were created in BioRender.com.

**Table 1 biomolecules-11-00466-t001:** *PRNP* resistance and susceptibility identified in animal prion diseases.

Disease	Species	*PRNP* Predisposition to Resistance		*PRNP* Predisposition to Susceptibility	
		Position	Amino acid	Position	Amino acid
Classical Scrapie	Sheep	112 136 137 141 142 154 171 171 176	T [12] A [13] T [11] F [19] K [11] H [13] K [20] R [13] K [11]	136 154 171	V [13] R [13] Q [13]
	Goat	143 146 146 154 211 222	R [21] D [22] S [22] H [13] Q [13] K [13]		
Atypical Scrapie	Sheep			141 154	F [13] H [13]
	Goat			154	H [23]
Chronic Wasting Disease	Elk	132	L [24]		
	White-tail deer	95 96	H [25] S [25]		
	Mule deer	225	F [26]		
Classical Bovine Spongiform Encephalopathy	Cattle			23-bp deletion (Promoter region) [27] 12-bp deletion (First intron) [28]	
Atypical Bovine Spongiform Encephalopathy				211	K [29]
				Indel 23-bp (Promoter region) [30] Indel 12-bp (First intron) [30]	
Camel Prion Disease	Dromedary camel	134E to be determined [31]		To be determined [32]	

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
