# Peer review of "Neuropathology of Animal Prion Diseases"

_biomolecules, 2021, doi:10.3390/biom11030466_

Round 1

Reviewer 1 Report

The manuscript requires proof-reading throughout for the numerous grammatical errors and some misleading sentences/language that were too numerous to list here.

  • Figure 1 – some of the images are quite dark – maybe they could be improved a little.
  • Throughout manuscript: PrPsc is generally written PrPSc and PrPc is PrPC
  • 5 the discussion of strains is not really clear – the writers have not defined what they mean by classical and atypical and the preceding definition of strains was also poorly communicated.
  • Figure 3. Not sure ‘natural’ is a good word to describe all the animal prion diseases shown here – were BSE and feline TSEs ‘natural’.
  • 6 Neuropathology not neuropatology?
  • 7 – rather than CS can be caused by several prion strains it would be better described “Several different prion strains have been associated with the development of classical scrapie” First identified in UK France Italy…
  • Figure 4 – presumably the brain sections are just illustrative of any animal brain as some are illustrating sheep, deer, cow prion densities/lesions on the same images. This should be clarified – in addition what each section represents as they are not the same section images for each model.
  • The description of neuroinflammation is poor considering the abundance of very recent studies. Many of the referenced works are from some years ago and this section could be very much improved by including recent works.  

Author Response

We would like to thank the reviewers for their reviews and recommendations

Page 3, Figure 1

We agree with the referee and have improved the quality of the images.

We agree with the referee and have made the suggested change throughout the manuscript: we now use PrPSc and PrPC , instead of PrPsc and PrPc, respectively.

Page 6, Lines 184 to 190

In order to clarify the definition of prion strains we add new text. Thus, it is now written “Different prion strains (isolates or variants) are distinct self-templating conformers [19] that display different phenotypes in a specified host and are preserved upon serial passage within the same host genotype. These distinct entities were first proposed in 1961 and since then have been characterized in different mammalian species with implication in disease pathology and transmission [20]. Prion strains can be differentiated after experimental transmission to wild type or transgenic rodents by differences in the clinical signs, incubation period and the lesion profiles in the brain of the affected animals leading to the laboratory definition of a strain [3].”

Page 7, Lines 214- 216

Concerning the terms “classical and atypical” we re-write the sentence: “As they showed distinct features from any TSE cases known at the time, the classical forms, they were designated as atypical (revised by [38]): atypical scrapie, atypical BSE and atypical CWD.”

Figure 3, Line 221

We agree with the reviewer and deleted the word “natural”. We would like to mention animal prion diseases in natural hosts regardless the type of transmission.

Page 8, Line 233

The word “3. Neuropathology” was corrected in the text.

Page 8, Line 248-249

As suggested by the reviewer, we changed the sentence to: “Several different prion strains have been associated with the development of classical scrapie. First identified in UK, France and Italy, those showed different biological features based on the serial passaging of natural isolates through bioassay experiments in rodents (reviewed by [37]).

Page 10, Figure 4

The brain sections are from sheep and are illustrative of the different neuroanatomical regions affected in animal prion diseases. To clarify, we include in Figure 4 the designation of each brain section.

Page 23, 4. Neuroinflammation

We agree with the reviewer that there are several recent works in this subject but they are most exclusively in experimental animal models. As this review is focused on neuropathology of animal prion diseases in natural hosts, the studies are more limited. Nevertheless, we included some new research studies hoping that these new data can be developed and analyzed in natural hosts.

Reviewer 2 Report

COMMENTS:

(1) Figure 1:

(A) Use arrows to mark relevant structures/features referred to in the legend and text (for example, vacuoles).

(B) Clarify the figure legend. There is no explanation as to what panels b and c represent.

(2) Figure 2:

Figures 2a and 2b are too small to read, at least as printed in the current pdf format provided for the review. Labels representing missense mutations in Figure 2b are especially difficult to see.

(3) It would be informative to tabulate the scientifically demonstrated susceptibility and resistance as a function of genetic predisposition, and including the corresponding literature references. The table columns could consist of: PrP disease, genetic predisposition to susceptibility (list mutations), genetic predisposition to resistance (list mutations), and references. Most of this information is scattered throughout the manuscript; tabulating it would increase the usefulness and value of this review.

(4) On pgs 9/33 and 16/33:

Formatting issue with listed items. Also, consider using bullet points instead of “-“.

(5) The “Neuroinflammation” section seems quite out of date, with only a few recent references. While it is true that there is much to be learned about the mechanisms involved in inflammation/prion disorders, this section of the manuscript should be updated.

(6) The manuscript would benefit from careful proofreading and addressing grammatical, punctuation, and stylistic errors.  Some examples include:

Pg. 21/33: Prion is defined as a “proteinaceous infectious particle” [6] consisting exclusively of a single protein without the involvement of nucleic acids that cause spongiform encephalopathies in mammals.

Pg. 20/33: The western blot characterization has shown that dromedary camels’ PrPsc from is further less glycosylated than those of CS, being characterized by a monoglycosylated dominant PrPsc and an apparent molecular weight slightly higher than CS and clearly higher than BSE and sheep passaged BSE [140].

Pg. 20/33: Immunohistochemical staining showed PrPsc deposition associated to vacuolation as well as in areas less or not affected by spongiosis such as the nucleus of the solitary tract, the hypoglossal nucleus; pyramidal cells of hippocampus; the granular layer of cerebellum, including Purkinje and several white matter areas.

Pg. 21/33: This altered phenotype may exacerbate the secretion of cytotoxic mediators and contribute to the spreading of prions, while increases the secretion of pro-inflammatory mediators by microglial cells [146].

Pg. 21/33: Pro-inflammatory cytokines and chemokines, among which IL-1α and β, IL-12p40, TNF, CCL2–CCL6, and CXCL10, are increased in the brains of mice with clinical disease) [142,150].

Author Response

We would like to thank the reviewers for their reviews and recommendations

  • Figure 1 (Page 3):
  • We agree with the suggestion of the reviewer and we introduced marks to point out the neuronal vacuolation (a), neuropil vacuolation (b), fine granular, stellate and perivascular (f), punctuate and globular labelling (i). In our opinion, the immunolabeling presented in the other images is self-explanatory.
  • The figure legend was clarified: “...(b) Neuropil vacuolation (arrow head) (Classical BSE, bovine, medulla oblongata, nucleus of the solitary tract); (c) Astrogliosis (Atypical scrapie, sheep, medulla oblongata, GFAP Polyclonal antibody, DAKO, 1:1000 dilution, x200);…”
  • Figure 2 (Page 4)

We agree with the reviewer and we improved both figures.

  • We agree with the reviewer and thank him for this suggestion. A table was introduced (Table 1) in page 5, gathering the known genetic susceptibility and resistance to prion diseases in animals.

We agree with the referee and have made the suggested changes in the text

  • We agree with the reviewer that there are several recent works in this subject but they are most exclusively in experimental animal models. As this review is focused on neuropathology of animal prion diseases in natural hosts, the studies are more limited. Nevertheless, we included some new research, hoping that these new data can be developed and analyzed in natural hosts.

  • To address the referees’ recommendations, we performed an accurate text revision throughout the manuscript. Several changes were made, namely in the sentences identified by the reviewer:
    • Page 3- “Prion is defined as a “proteinaceous infectious particle” [6] consisting exclusively of a single protein without the involvement of nucleic acids that causes spongiform encephalopathies in mammals.”
  • Page 23- “Immunohistochemical staining showed PrPSc deposition associated to vacuolation but also in areas less or not affected by spongiosis such as the nucleus of the solitary tract, the hypoglossal nucleus, the pyramidal cells of hippocampus, the granular layer of cerebellum, the Purkinje cells and several white matter areas.”
  • Page 23- “The western blot characterization has shown that dromedary camels’ PrPSc is less glycosylated than those of CS. It presents a monoglycosylated dominant PrPSc and an apparent molecular weight slightly higher than CS and clearly higher than BSE and sheep passaged BSE [32]”;
  • Page 24- “This altered phenotype may exacerbate the secretion of cytotoxic mediators and contributes to the spreading of prions, while increases the secretion of pro-inflammatory mediators by microglial cells [161].”
  • Page 24- “Pro-inflammatory cytokines and chemokines, among which IL-1α and β, IL-12p40, TNF, CCL2–CCL6, and CXCL10, are increased in the brains of mice with clinical disease [1156, 167]”.